# Causal effect of children's secondary education on parental health outcomes: findings from a natural experiment in Botswana

Jan Ole Ludwig,[1] Neil M Davies,[2,3,4] Jacob Bor,[5,6] Jan-Walter De Neve [iD] [1]

For numbered affiliations see end of article.

**Correspondence to**
Dr Jan-Walter De Neve;
janwalter.deneve@uni-heidelberg.de

## ABSTRACT

**Objectives** A growing literature highlights the intergenerational transmission of human capital from parents to children. However, far less is known about 'upward transmission' from children to parents. In this study, we use a 1996 Botswana education policy reform as a natural experiment to identify the causal effect of children's secondary schooling on their parents' health.

**Setting** Botswana's decennial census (2001 and 2011). Data were obtained through the Integrated Public Use Microdata Series and are 10% random samples of the complete population in each of these census years.

**Participants** Survey respondents who were citizens born in Botswana, at least 18 years old at the time of the census and born in or after 1975 (n=89 721).

**Primary and secondary outcome measures** Parental survival and disability at the time of the census, separately for mothers and fathers.

**Results** The 1996 reform caused a large increase in grade 10 enrolment, inducing an additional 0.4 years of schooling for the first cohorts affected (95% CI 0.3 to 0.5, p<0.001). The reform, however, had no effect on parental survival and disability by the time exposed child cohorts reach age 30. Results were robust to a wide array of sensitivity analyses.

**Conclusions** This study found little evidence that parents' survival and disability were affected by their offspring's educational attainment in Botswana. Parents' health may not be necessarily affected by increasing their offspring's educational attainment.

## Strengths and limitations of this study

- ► While a large literature has investigated the role of parental human capital on children's well-being, relatively little is known about 'upward transmission' from children to parents.
- ► We explore the 1996 Botswana education policy reform as a natural experiment to estimate the causal effect of children's secondary education on their parents' health.
- ► Using nearly 90 000 census records from 2001 and 2011, we show that the 1996 reform resulted in an additional 0.4 years of schooling among exposed child cohorts.
- ► Children's education, however, had few detectable effects on parental survival and disability. Parents' health may not be affected by increasing their offspring's education.
- ► Limitations of our study include limited generalisability beyond the reform analysed and lack of information on pathways and alternative parental health outcomes.

## INTRODUCTION

Investments in children's human capital may yield large dividends for themselves[1], their offspring,[2,3] and for their parents. While a large literature has documented the substantial resource flows from parents to children, a growing literature from Asia,[4] Latin America[5] and sub-Saharan Africa[6] suggests that intergenerational wealth transfers have started to reverse in ageing low and middle-income societies where governmental support for older adults is limited.[7] In addition to sharing their wealth with parents, better educated children may share their increased knowledge[8]

and skills[9] with older household members, and may be better equipped to interact with the health system to help manage the health needs of their ageing parents. Additional education may also open access to careers in the health sector, leading to increased intra-family medical expertise, with positive health benefits for family members.[10] Adult children have been suggested to 'repay' their parents for investments received in childhood, particularly in settings where families function as de facto social institutions.[11]

The past decades have brought a massive increase in educational opportunities for younger cohorts in low and middle-income countries (LMICs). Gross domestic product per capita in LMICs more than doubled between 2000 and 2018, from $2310 to $4662 in 2010 US$.[12] The extent to which this younger generation shares its increased resources with their parents, however,

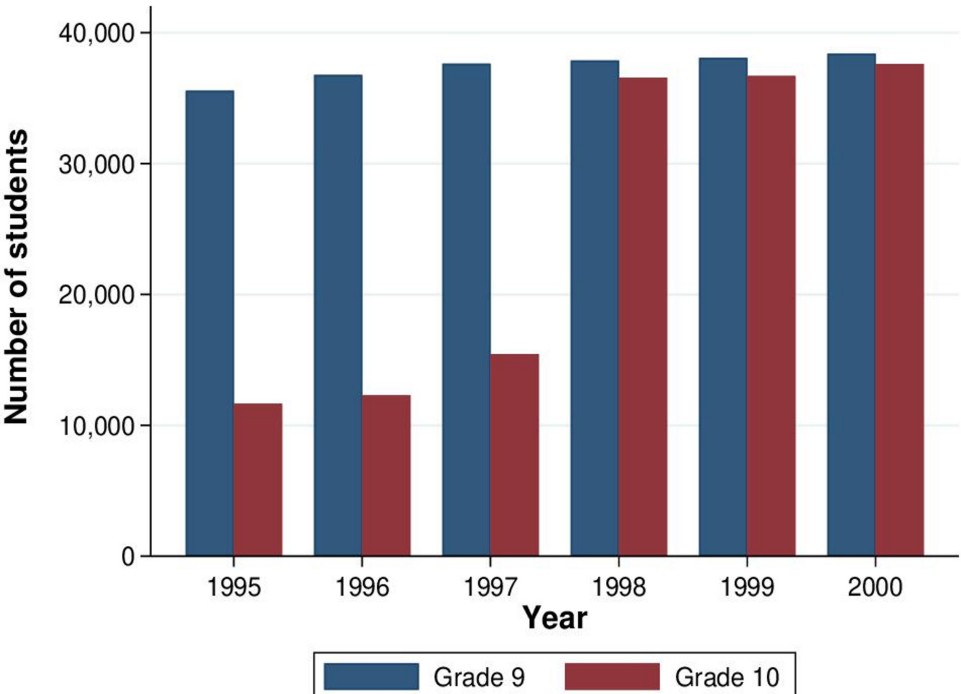

**Figure 1** Enrolment in grades 9 and 10 in Botswana. This figure shows the total enrolment in grades 9 and 10 in Botswana by year. In January 1996, Botswana shifted grade 10 from senior secondary to junior secondary school, with the goal of increasing access. Botswana's education system is highly centralised and the reform was implemented rapidly throughout the country. More teachers were hired and classrooms were created. Moving grade 10 to junior secondary reduced travel time for students in addition to expanding the number of grade 10 seats. The reform also raised the benefits of completing grade 10 because it is now required to obtain a Junior Certificate. The result of this reform was a large increase in the number of students attending grade 10 from 1997 to 1998. Source: Botswana Education Statistics 1995–2000.

remains largely unknown. More than a dozen studies have reported a positive correlation between children's human capital and parental health, including in China,[13] India[14] and South Africa.[15] Given the large number of potential factors (such as genetic traits) that may confound the relationship between children's human capital and parental health, however, these correlations are difficult to interpret and are unlikely to identify causal effects of educational opportunities provided to children. A recent review concluded that spillover effects from offspring to parents are understudied and (quasi)experimental studies from low-resource settings are needed.[16]

In this study, we explore a large secondary schooling reform in sub-Saharan Africa to determine the impact of increasing educational investment in younger generations on their parents. In 1996, the government of Botswana launched a major education reform. Between 1997 and 1998, grade 10 enrolment in the country more than doubled from 15 437 to 36 562 (figure 1; see online supplemental table 1 and online supplemental material 1 for additional background). At the cohort level, the reform induced an additional 0.4 years of schooling among affected cohorts relative to the attainment the affected cohorts would have achieved if previous trends had continued. This reform constitutes a 'natural experiment' through comparison of cohorts exposed to the reform versus those unexposed, while controlling for gradual changes across cohorts. Using two waves of

Botswana's decennial census (2001, 2011) to disentangle age and cohort effects, we used the resulting variation in exposure to identify the causal effect of children's secondary education on parental health.

## MATERIALS AND METHODS
### Data sources and study population
Data for this study were obtained from the Botswana Population and Housing Census 2001 and 2011 through the Integrated Public Use Microdata Series (IPUMS).[17] The microdata provided by IPUMS are 10% random samples of the complete population in each of these years (ie, 'full censuses'). Subnational administrative boundaries were harmonised across censuses by IPUMS. The censuses were conducted by the Central Statistics Office Botswana and included all persons in the country at the time of the census without regard of their usual residence. The censuses also collected basic demographic information on members of the household who were citizens outside of the country on the census night, but who would typically live in the household if they would have been in Botswana. In the 2011 census, for instance, 24 561 citizens were living outside of Botswana (1.2%). Data on sex, age, years of schooling and the survival status of mothers and fathers were available for 88.9% of respondents, yielding a total of 333 334 individuals in the pooled 2001 and 2011

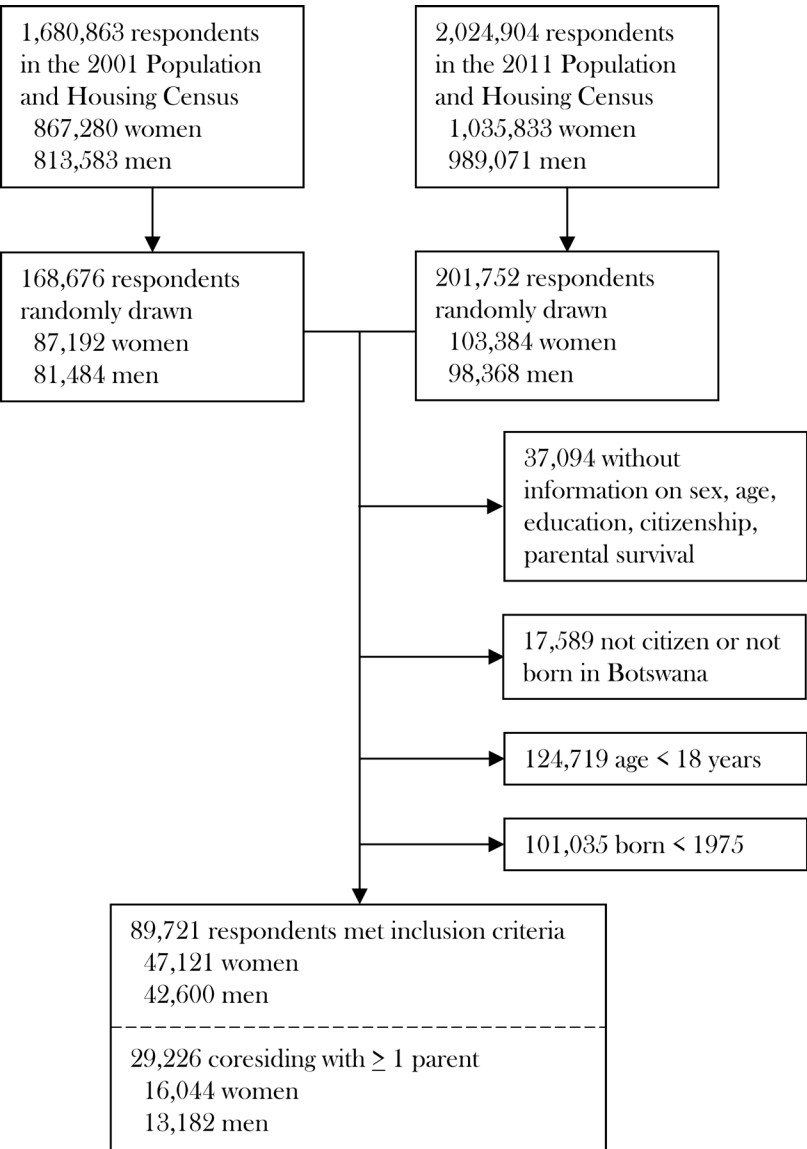

**Figure 2** Study participants. This figure shows the flow of participants through the Botswana Census (2001 and 2011).

census data sets (figure 2). The census data do not allow matching individuals between census years.

The study population included all citizens of Botswana and born in Botswana. Children aged younger than 18 were excluded because they would not have had the opportunity to complete secondary school. Children born prior to 1975 were excluded because previous secondary school reforms led to changes in education for these older cohorts. Using a narrow subsample of pre-form and postreform cohorts (ie, born between 1975 and 1993 based on the age and birth cohort cut-offs described above), there were good reasons to believe that control cohorts differed from exposed cohorts mostly with respect to their access to grade 10. We explore alternative sample specifications in sensitivity analyses (Sensitivity Analyses section). Children not born in Botswana were excluded because they would not have been exposed to the reform if they migrated in adulthood.

### Measurement of exposure and endpoints

Data on education, age in years, gender, citizenship, district of birth, parental survival, parental disability, parental age and parental years of schooling were extracted from the census data sets. Year of birth was calculated based on age collected in the census. The key independent variable of interest was the child's total years of schooling ('What is the highest grade/level that [respondent] has completed?'). Only formal schooling was counted. Years of school account for the number of years of study regardless of the track or kind of study. Our main outcome of interest was parental survival status, separately for mothers and fathers. Parental survival status indicated whether the person's biological parent was alive at the time of the census (eg, 'Is [the person's] biological mother alive?') and was observed regardless of whether the parent is coresident with the adult child. During the period of study, Botswana had very high adult mortality

rates due in large part to the country's HIV epidemic. In 2001, the probability of a 15 year-old reaching age 60 ($_{45}e_{15}$) was about 37%.[18] In this context, children's education could affect parental survival because information about HIV was disseminated via secondary school 'life skills' curricula[19]; because HIV stigma may be lower among the better educated, leading to higher HIV testing[20]; and/or because better educated children may help their parents navigate the health system and gain access to HIV treatment.[9] Other major determinants of morbidity and mortality among adults aged 50–69 years in Botswana include cardiovascular diseases, neoplasms, and diabetes and kidney diseases.[21] Diabetic retinopathy and peripheral neuropathy, for instance, are increasingly common causes of severe visual impairment and non-traumatic lower extremity disability, including in the context of sub-Saharan Africa.[22]

Pointer variables provided by IPUMS indicate the location within the household of every child's parent, and make it possible to construct individual-level variables representing the characteristics of coresident parents. For the subset of households with surviving, coresident parents and children, we were therefore able to assess the impact of the schooling reform on key parental socioeconomic characteristics and disability as additional secondary outcomes. Children's education, for instance, has been suggested to affect parental fertility decisions by creating additional costs (educational expenses) and decreasing the perceived benefits derived from children by removing them from the family productive system.[23] The census also asked a series of questions about disability in the 2001 and 2011 waves. Specifically, they asked about employment disability, blind or vision impaired, deaf or hearing impaired, mute or speech impaired, disability affecting lower extremities, disability affecting upper extremities, mental disability and psychological disability. For our analysis, we coded the following parental disability variables: 'employment disability' defined as whether a parent was economically inactive because of health-related reasons; 'blind or vision-impaired' defined as whether a parent was blind or had limited vision; and 'disability affecting lower extremities' defined as whether a parent was unable to use one or both legs.[17] Disabilities were reported by a household respondent (typically the household head). Data on financial transfers between children and their parents, frequency of contact between children and parents and the uptake of social or healthcare services were not collected during the census.

## Unit of analysis

Our unit of analysis was the adult child. In doing so, we follow the data structure and variables provided by IPUMS closely, which maps well unto our analytical approach since our identification strategy is "based on cohort-specific variation to the education reform at the child level".[24] Moreover, the census data sets do not contain information on all offspring's educational attainment which precludes an analysis at the parent level.[17] The

census data sets thus differ from more standard surveys focused on older adults and which contain information on the educational attainment of all offspring (eg, the Health and Retirement Study in the USA). For our main analysis, we included adult children regardless of whether they were the head of household, their marital status or whether the child coresided with their parent(s).

## Instrumental variable analysis

An instrumental variable (IV) is a variable that is associated with the treatment of interest, but is independent of confounding factors and has no direct effect on the outcome.[25] In our application, the instrument was a 'reform cohort' indicator, taking the value 1 if the respondent was born in a cohort that was exposed to the education policy reform and 0 otherwise. In Botswana, children were historically expected to start primary school at age 7 years and enter junior secondary school at age 15 years. Individuals born in 1981 or later would therefore have entered junior secondary school in 1996 or later and were classified as exposed (online supplemental figure 1). We used this exogenous variation in educational attainment resulting from the policy reform to identify the causal effect of children's education on their parents' survival. First, we assessed whether children's cohorts exposed to the reform had higher educational attainment than birth cohorts not exposed to the reform using multivariable adjusted ordinary least squares (OLS) linear probability regression models. We estimated the effect of exposure to the reform on total years of schooling completed and the probability of completing at least 10 years of schooling. Second, as a benchmark for our causal analysis, we estimated the 'naïve' cross-sectional association of children's education with their parents' survival, acknowledging that this association may reflect reverse causality and effects of confounders and therefore does not have a causal interpretation. We assessed the crude relationship graphically and in multivariable adjusted regression analyses. Third, we assessed the intention-to-treat (ITT) effect of having a child affected by the reform on parental survival. We examined the relationship graphically and adjusted for covariates. Fourth, we estimated two-stage least squares (2SLS) (IVs) regression models using exposure to the reform as an instrument for total years of children's schooling while adjusting for covariates. Natural experiments that change the probability of an exposure can be analysed like randomised controlled trials with non-compliance. Exposure to the reform was modelled as an intercept shift for cohorts affected by the reform.

In all models, we controlled for children's single-year age indicators to account for trends in age and education across children's cohorts. We focused on this age specification as it is highly flexible, but assess alternate specifications in sensitivity analyses. We also included a linear term in children's year of birth and indicators for children's district of birth. We estimated models for women and men separately as well as a pooled sample. When pooling sexes, we additionally included children's sex and the

interactions of sex with all other covariates. Data on parent's own sociodemographic characteristics were also available for the subsample of children coresiding with at least one parent. When testing the association between children's education and secondary parental outcomes in 'conventional' multivariable regression analyses, we therefore additionally included controls for parental age (years) and parental education (years of schooling).

## Assumptions for causal inference

For a causal interpretation of our effect estimates, four assumptions must be satisfied (see online supplemental figure 2 for a directed acyclic graph). First, our instrument must be associated with educational attainment. This is testable and we found evidence of large effects (F-statistic: 65.6). The large value of the F-statistic indicates that our instrument was strongly associated with schooling. Second, the instrument must be independent of unobserved confounders of the instrument–outcome association conditional on observed covariates. This implies that children born before and after the reform cut-off were similar after controlling for children's single-year age indicators, a continuous term in children's year of birth and children's district of birth. Critically, the availability of multiple census years allowed us to generate variation in children's age in a given birth cohort group. Our models are therefore also robust to the existence of period effects, which we control for by simultaneously adjusting for children's age and a continuous term in year of birth (noting that census_year=birth cohort+age). We also assessed alternative specifications of children's age (quadratic, cubic, quartic) and year of birth (quadratic), and restricted the sample to a narrower birth cohort window around the reform cut-off (±3 years and ±5 years around the cut-off). We also allowed for different linear slopes in year of birth on either side of the reform cut-off akin to a 'fuzzy' regression discontinuity design.[26]

We also used covariate balance plots to estimate the association of our exposure (schooling) and our instrument (reform indicator) with observed covariates, as described in detail elsewhere.[27 28] While this analysis cannot prove that our instrument is not associated with unobserved covariates, it can provide some indication if our instrument is less strongly associated with measured covariates than our exposure. Third, for our 2SLS estimates to have a causal interpretation, we assume that exposure to the policy reform affected our parental outcomes only through changes in secondary schooling. This assumption is plausible given that the education reform was a supply-side intervention,[19] which would not have affected parents except through their children's increased access to school. Fourth, to interpret our results as local average treatment effects, we assume that there are no defiers (ie, the monotonicity assumption).[29]

## Sensitivity analyses

In addition to the robustness checks described above, we conducted a wide range of sensitivity analyses, including

earlier birth cohorts (ie, those born between 1971 and 1993, instead of those born between 1975 and 1993), modelling the outcome using a logit link function and controlling for an indicator for heap year (defined as ages that are a multiple of 5) to account for measurement error in age due to potential age heaping. Results could also be biased by differential response rates by birth cohort, manipulation of our 'running variable', or by selection bias from mortality risk or migration associated with being born after 1980. We therefore assessed cohort sizes by birth cohort (online supplemental figure 3). We note that we present our results for secondary parental outcomes under the null hypothesis that there is no effect of the reform on parental survival.

To further probe the robustness of our main findings, we conducted a number of subsample analyses. Specifically, we included households for whom additional education among children was most likely to have relatively large consequences for parental survival. Our hypothesis was that the role of children's resources may be more pronounced in households where: (1) children were more likely to be affected by the reform, (2) there was a large intergenerational gap in educational attainment between adult children and their parents, and (3) parents were on average older and thus more likely to be 'dependent' on the support of children.[30] To do so, we limited the sample to children with at least 9 years of schooling (ie, those primarily affected by the policy reform). For the subset of households with coresident parents and children, we also limited the sample to households where the difference in educational attainment between parents and children was ≥5.5 years (50th percentile), and stratified our model by parental 10-year age groups.

## Data and code availability

This was a complete case analysis and all analyses were conducted in StataMP V.15.1. Data are publicly available at no cost from IPUMS (http://ipums.org).[17] Code used to generate these results can be found at https://github.com/jwdeneve/Botswana-ParentalHealth.

## Patient and public involvement

Patients or the public were not involved in the design, or conduct, or reporting, or dissemination plans of our research.

## RESULTS

### Descriptive statistics

On average the children in our analytical sample were 24.8 years old (table 1). Across 34 202 children born prereform, 13 073 (38.2%) had lost their father and 5666 (16.6%) had lost their mother. In terms of household characteristics, the average number of family members per household was 5.0. In our subset of households with surviving, coresident parents and children, there were a total of 29 266 (32.6%) child–parent dyads (10 082 of which were living with both parents). Among those, the

**Table 1** Selected characteristics of estimation sample

| | Prereform cohorts (n=34 202) | | Postreform cohorts (n=55 519) | |
|---|---|---|---|---|
| | Census 2001 | Census 2011 | Census 2001 | Census 2011 |
| **Children's characteristics** | | | | |
| Mother alive (%) | 88.5 | 77.5 | 90.9 | 81.1 |
| Father alive (%) | 71.3 | 50.7 | 77.6 | 63.2 |
| Age, mean (SD) | 23.4 (1.7) | 33.4 (1.7) | 18.9 (0.8) | 23.8 (3.8) |
| Years of schooling, mean (SD) | 8.9 (3.4) | 9.5 (4.0) | 9.3 (3.0) | 10.4 (3.3) |
| Has at least 10 years of schooling (%) | 42.0 | 45.3 | 65.1 | 77.1 |
| Labour force participation (%) | 65.3 | 83.6 | 37.3 | 65.0 |
| Speaks Setswana at home (%) | 82.5 | 83.8 | 82.0 | 81.9 |
| Christian (%) | 70.2 | 79.0 | 71.8 | 78.0 |
| Household characteristics | | | | |
| Number of family members in household, mean (SD) | 5.2 (3.9) | 4.5 (3.5) | 6.1 (3.8) | 4.8 (3.7) |
| Any deaths in household last year (%) | 5.7 | 2.7 | 6.4 | 2.9 |
| Ownership of dwelling (%) | 63.0 | 57.1 | 74.6 | 62.4 |
| Access to electricity (%) | 24.6 | 57.2 | 23.4 | 55.4 |
| Access to piped water (%) | 19.5 | 31.0 | 18.5 | 28.2 |
| Telephone availability (%) | 39.1 | 10.4 | 36.6 | 11.4 |
| Cellular phone availability (%) | – | 94.2 | – | 93.9 |
| Receives remittances (%) | 21.5 | 30.4 | 23.1 | 32.1 |

Sample includes survey respondents who were citizens of Botswana, born in Botswana, at least 18 years old at the time of the census and born in or after 1975. The unit of analysis was children born either during the prereform period (prior to 1981) or postreform period (in or after 1981). Source: Botswana Census 2001 and 2011.

average parental age was 50.8 and 57.0 years for mothers and fathers, respectively. The average gap in attained education between children and coresiding parents was substantial (5.2 and 6.0 years of schooling for mothers and fathers, respectively). Online supplemental table 2 shows detailed descriptive statistics for the subset of children coresiding with at least one parent.

### Association between children's education and parental survival

Figure 3 illustrates heterogeneity in the unadjusted relationship between children's education and parental survival, separately for maternal and paternal survival. The positive association between children's education and parental survival was non-linear, levelling off after junior secondary school. In 'conventional' multivariable adjusted regression analyses, children's schooling was associated with increased parental survival. In the pooled (daughters and sons) sample, each additional year of children's schooling was associated with a 0.8 percentage point increase (95% CI 0.7 to 0.9) and a 0.6 percentage point increase (95% CI 0.5 to 0.7) in the probability of maternal and paternal survival, respectively. These associations persisted after controlling for parent's own sociodemographic characteristics in the subset of households with coresident parents and children (online supplemental table 3). Although suggestive, these associations

may be vulnerable to reverse causality or confounding by unobserved characteristics.

### Causal effect of children's education on parental survival

Table 2 shows the estimated reform impact on children's educational attainment, as indicated by total years of schooling completed and the probability of having completed at least 10 years of schooling. All models controlled flexibly for single-year age indicators, a continuous term in year of birth, indicators for district of birth and census year. In the pooled sample, the estimated effect of the reform on educational attainment was 0.4 additional years of schooling (95% CI 0.3 to 0.5) among exposed child cohorts. Exposure to the reform was also associated with a 19.0 percentage point increase (95% CI 17.5 to 20.4) in the probability of having completed at least 10 years of schooling relative to a baseline probability of 43.5% among prereform child cohorts.

Online supplemental figure 4 plots parental survival and disability against reform exposure. We found little evidence of a discontinuity for postreform cohorts. Multivariable regression models—which adjusted for children's single-year age indicators, a continuous trend in children's year of birth, district of birth indicators and census year indicators—presented in table 3 also provide little evidence (at alpha=0.05, with 80% power) that exposure to the reform affected maternal survival (−0.2 percentage

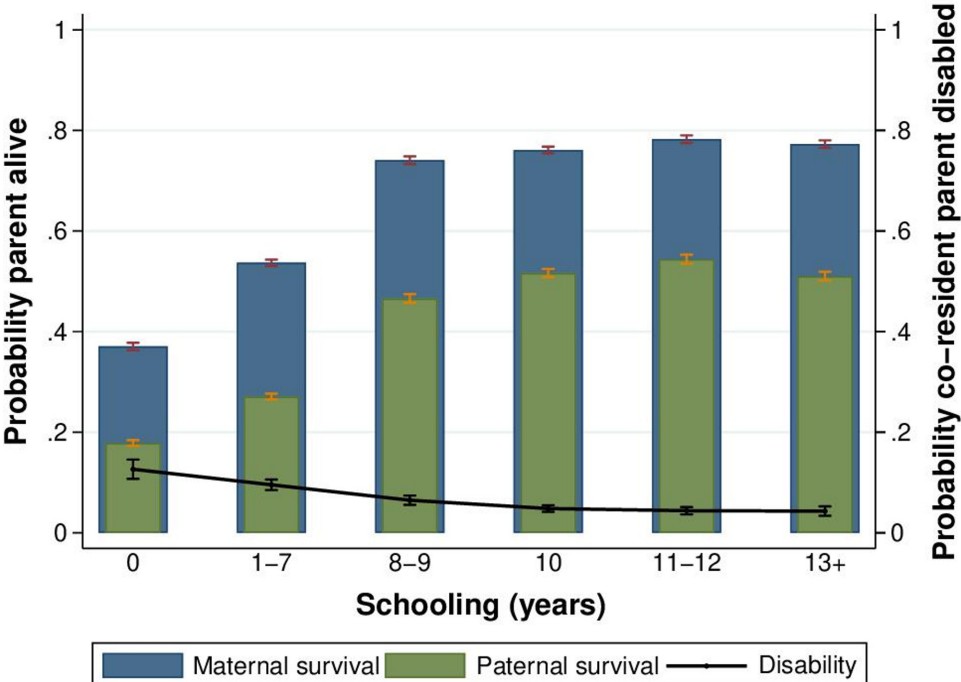

**Figure 3** Parental survival and disability by children's educational attainment. This figure shows the unadjusted relationship between children's schooling and parental survival status and disability. Educational attainment was defined as total years of schooling completed by the time of the census. Disability was defined as the probability that any co-resident parent was disabled. Strong protective effects are seen for primary school completion, with effects levelling off after junior secondary school. The sample for parental survival includes respondents who are Botswana citizens, born in Botswana and aged at least 22 at the time of the census. The sample for parental disability includes respondents who are Botswana citizens, born in Botswana, aged at least 22 and living with at least one parent at the time of the census. Source: Botswana Census 2011.

| Table 2 First stage results: effect of the reform on children's educational attainment | | | |
|---|---|---|---|
| **Subsample** | **Daughters** | **Sons** | **Both sexes** |
| **Dependent variable** | | | |
| Child's schooling (years) | **Difference in years of schooling (95% CI)** | | |
| Reform indicator | 0.458* | 0.365* | 0.414* |
| | (0.330 to 0.586) | (0.210 to 0.521) | (0.314 to 0.514) |
| Observations | 47 121 | 42 600 | 89 721 |
| R² | 0.088 | 0.080 | 0.088 |
| Child has ≥10 years of schooling (1=yes, 0=no) | **Difference in Pr(Educ≥10) (95% CI)** | | |
| Reform indicator | 22.3* | 15.4* | 19.0* |
| | (20.3 to 24.2) | (13.3 to 17.5) | (17.5 to 20.4) |
| Observations | 47 121 | 42 600 | 89 721 |
| R² | 0.152 | 0.118 | 0.136 |

Regression results from ordinary least squares models controlling for children's single-year age indicators, a continuous trend in children's year of birth and children's district of birth. Regressions for the subsample with both children's sexes additionally control for an indicator for children's sex and interactions of each covariate with children's sex. Our models are robust to period effects, which we controlled for implicitly by simultaneously adjusting for children's age and year of birth. Children's exposure to the reform was defined as a binary 'Reform indicator' (1=year of birth>1980; 0=otherwise). Pr(Educ≥10) was defined at the probability of a child completing at least 10 years of schooling by the time of the census (1=yes, 0=no). The sample includes survey respondents who were citizens born in Botswana, at least 18 years old at the time of the census and born in or after 1975. 95% robust CIs in parentheses. Binary outcomes were multiplied by 100 to facilitate the interpretation of coefficients and SEs on a % point scale. Source: Botswana Census 2001 and 2011.
*P<0.01

**Table 3** The association of children's schooling and parental survival

| Dependent variable | Mother alive (1=yes, 0=no) | | | Father alive (1=yes, 0=no) | | |
|---|---|---|---|---|---|---|
| Subsample | Daughters | Sons | Both sexes | Daughters | Sons | Both sexes |
| Risk difference (95% CI) | | | | | | |
| A. OLS model | | | | | | |
| Child's schooling (years) | 0.9* | 0.7* | 0.8* | 0.8* | 0.5* | 0.6* |
| | (0.8 to 1.0) | (0.6 to 0.8) | (0.7 to 0.9) | (0.6 to 0.9) | (0.4 to 0.6) | (0.5 to 0.7) |
| B. OLS model | | | | | | |
| Child has ≥10 years of schooling (1=yes, 0=no) | 4.6* | 3.7* | 4.1* | 4.0* | 1.9* | 3.0* |
| | (3.8 to 5.4) | (2.9 to 4.5) | (3.6 to 4.7) | (3.0 to 4.9) | (0.9 to 2.9) | (2.3 to 3.7) |
| C. ITT model | | | | | | |
| Reform indicator | −0.5 | 0.2 | −0.2 | 0.4 | 0.7 | 0.5 |
| | (−2.1 to 1.0) | (−1.4 to 1.8) | (−1.3 to 0.9) | (−1.6 to 2.5) | (−1.5 to 2.8) | (−0.9 to 2.0) |
| D. 2SLS model | | | | | | |
| Child's schooling (years) | −1.2 | 0.6 | −0.4 | 0.9 | 1.8 | 1.3 |
| | (−4.6 to 2.3) | (−3.9 to 5.0) | (−3.2 to 2.3) | (−3.5 to 5.4) | (−4.1 to 7.7) | (−2.3 to 4.9) |
| F-statistic | 49.0 | 21.1 | 65.6 | 49.0 | 21.1 | 65.6 |
| Dependent variable, prereform (%) | 83.3 | 83.4 | 83.4 | 61.3 | 62.4 | 61.8 |
| Observations | 47 121 | 42 600 | 89 721 | 47 121 | 42 600 | 89 721 |

Panels A and B show regression results from an ordinary least squares (OLS) model controlling for children's single-year age indicators, a continuous trend in children's year of birth and children's district of birth. Regressions for the subsample with both children's sexes additionally control for an indicator for children's sex and interactions of each covariate with children's sex. Our models are robust to period effects, which we controlled for implicitly by simultaneously adjusting for children's age and year of birth. 'Child has ≥10 years of schooling' was defined at the probability of a child completing at least 10 years of schooling by the time of the census (1=yes, 0=no). Panel C shows regression results from an intention-to-treat (ITT) OLS model in which children's exposure to the reform was defined as a binary 'Reform indicator' (1=year of birth>1980; 0=otherwise). Panel D shows regression results from a two-stage least squares (2SLS) model, in which exposure to the reform was used as an instrument for children's years of schooling. Sample includes survey respondents who were citizens born in Botswana, at least 18 years old at the time of the census and born in or after 1975. Binary outcomes were multiplied by 100 to facilitate the interpretation of coefficients and SEs on a % point scale. 95% robust CIs in parentheses. Source: Botswana Census 2001 and 2011.
*P<0.01

points; 95% CI −1.3 to 0.9) or paternal survival (0.5 percentage points; 95% CI −0.9 to 2.0). Our results were generally not sensitive to different modelling strategies for age, different specifications of the outcome, sample inclusion criteria (online supplemental tables 3–6) or model (online supplemental table 7). The differences in potential confounders for the actual exposure (schooling) and the proposed instrument (reform cohort) are presented in online supplemental figure 5. These results suggest that the IV analysis would be generally less biased from these measured potential confounders than the multivariable adjusted regression analysis. One variable appeared to differ for the instrument (maternal schooling); however, adjusting for this variable did not meaningfully affect our results.

### Children's education and secondary parental outcomes
In table 4, we show results for our secondary parental outcomes, including key parental demographic and socioeconomic characteristics and disability categories. Consistent with our main results for maternal and paternal survival, we found little evidence that children's

exposure to the reform affected any of these secondary parental outcomes (panel C, table 4).

### DISCUSSION
Using census data from 89 721 adult children, we find that additional secondary education in children had few detectable survival benefits for their parents in Botswana. Although children's education was strongly associated with improved parental health in OLS regression models, these results could be due to, for example, reverse causality whereby chronically ill parents require care from children before completion of school. Our ITT estimates suggest that it is unlikely that children's exposure to the reform increases maternal and paternal survival by more than 0.9 and 2.0 percentage points, respectively, which corresponds to the value of the upper limit of the 95% CI. Our 2SLS estimates suggest that it is unlikely that each additional year of schooling increases maternal and paternal survival by more than 2.3 and 4.9 percentage points, respectively. Effects were consistent across a wide array of robustness checks, including alternative specifications of

**Table 4** The association of children's schooling and secondary parental outcomes

| Dependent variable<br><br>Sample: both sexes | Number of children ever born to mother | Age of mother's youngest child in household (years) | Maternal labour force participation (1=yes, 0=no) | Parental employment disability (1=yes, 0=no) | Parental blindness (1=yes, 0=no) | Parental lower extremity disability (1=yes, 0=no) |
|---|---|---|---|---|---|---|
| **A. OLS model** | | | | | | |
| Child's schooling (years) | −0.2* | 0.1* | 1.6* | −0.5* | −0.1† | −0.1* |
| | (−0.2 to −0.1) | (0.0 to 0.1) | (1.4 to 1.7) | (−0.6 to −0.4) | (−0.2 to −0.0) | (−0.1 to −0.0) |
| **B. OLS model** | | | | | | |
| Child has ≥10 years of schooling (1=yes, 0=no) | −0.7* | 0.3* | 8.5* | −2.2* | −0.5† | −0.5* |
| | (−0.8 to −0.6) | (0.1 to 0.5) | (7.2 to 9.7) | (−2.8 to −1.6) | (−0.9 to −0.0) | (−0.8 to −0.2) |
| **C. ITT model** | | | | | | |
| Reform indicator | 0.0 | 0.3 | −2.1 | 0.3 | 0.0 | 0.1 |
| | (−0.1 to 0.1) | (−0.1 to 0.7) | (−4.5 to 0.7) | (−0.7 to 1.4) | (−0.8 to 0.8) | (−0.4 to 0.7) |
| **D. 2SLS model** | | | | | | |
| Child's schooling (years) | 0.0 | 0.7 | −4.8 | 0.8 | 0.0 | 0.3 |
| | (−0.3 to 0.3) | (−0.2 to 1.6) | (−10.5 to 2.0) | (−1.5 to 3.1) | (−1.8 to 1.8) | (−0.8 to 1.5) |
| F-statistic | 29.8 | 29.8 | 29.8 | 33.4 | 33.4 | 33.4 |
| Probability dependent variable, prereform | 6.3 | 17.8 | 38.8 | 6.3 | 3.8 | 1.6 |
| Observations | 26 664 | 26 664 | 26 664 | 29 212 | 29 212 | 29 212 |

Panels A and B show regression results from an ordinary least squares (OLS) model controlling for children's single-year age indicators, a continuous trend in children's year of birth, children's district of birth, an indicator for children's sex and the interactions of each covariate with children's sex. Our models are robust to period effects, which we controlled for implicitly by simultaneously adjusting for children's age and year of birth. 'Child has ≥10 years of schooling' was defined at the probability of a child completing at least 10 years of schooling by the time of the census (1=yes, 0=no). Panel C shows regression results from an intention-to-treat (ITT) OLS model in which exposure to the reform was defined as a binary indicator (1=year of birth>1980; 0=otherwise). Panel D shows regression results from a two-stage least squares (2SLS) model, in which exposure to the reform was used as an instrument for years of schooling. Sample includes survey respondents who were citizens born in Botswana, at least 18 years old at the time of the census, born in or after 1975 and coresided with their mother (columns 2–4) or at least one parent (columns 5–7) at the time of the census. Binary outcomes were multiplied by 100 to facilitate the interpretation of coefficients and SEs on a % point scale. 95% robust CIs in parentheses. Source: Botswana Census 2001 and 2011.
*P<0.01
†P<0.05

long-run trends, analytical sample and multiple parental outcomes. As with all instruments, a number of assumptions underpin a causal interpretation of these findings. A key assumption underlying our model is that the timing of children's birth around the reform period is random conditional on covariates (no confounding assumption). We controlled for generic time trends in our multivariable models to absorb gradual changes in child and parental characteristics over time. The main logic underlying our identification strategy is that the reform led to an unanticipated *additional* increase in children's schooling among affected cohorts.

There are many reasons why parental socioeconomic characteristics and health outcomes might change across children's birth cohorts. Notably, in the early 2000s, Botswana became the first country in sub-Saharan Africa to offer large-scale public treatment for HIV to all qualifying citizens, which had massive and immediate effects on economic and health outcomes.[31] Botswana

has also implemented some of the most successful drought relief programmes in Southern Africa, which were implemented to address the loss of livestock and improve nutritional outcomes.[32] These changes in public health prevention programming and socioeconomic context in Botswana, however, tend to be gradual over time and/or affect children of different ages. These phenomena would result in gradual changes in parental survival across children's ages and would be picked up in our controls for underlying trends in children's age. The availability of multiple census years also allowed us to generate variation in children's age in a given birth cohort. Our models were therefore robust to period effects (eg, differences in HIV mortality and health systems across census years), which we controlled for implicitly by simultaneously adjusting for children's age and year of birth. Although violations of these IV conditions appear unlikely in our study, it is impossible to prove that they are definitively met.

While the educational reform likely allows us to identify the causal effects of interest, relying on changes generated by the 1996 education reform and census data comes with limitations in terms of external validity. First, all of the variation explored reflects variation in secondary schooling. Effects of schooling may be qualitatively and quantitatively different in primary school[24 33] or higher education,[10] and indeed the association between measured schooling and parental survival was non-linear. Second, the causal effects we identify do not necessarily represent the causal effects in the general population but in the subpopulation of compliers (ie, children who attained increased schooling because of the reform). Nevertheless, one advantage of using Botswana's supply-side school reform is the important size of its impact (nearly half a year of additional schooling on a population level), and that it affected a large segment of the population. Third, the opportunity costs of schooling (eg, not being able to work) are part and parcel of schooling, and it is impossible to increase schooling without reducing whatever activities children would have been doing at the time. Therefore, the exposure could be most precisely stated as an additional year of schooling—and a year less of whatever the child would have been doing otherwise. Fourth, educational attainment reported in the census may be subject to measurement error, which may bias the multivariable adjusted estimates towards the null. The IV analysis, however, is robust to classical (balanced) measurement error in education. Fifth, our measures of parental health may be susceptible to reporting bias. Self-reported health, however, has been suggested to be a valid and reliable indicator when assessing the health of adults in low and middle-income settings.[34] Sixth, parental disability outcomes which are available in the census data, such as visual impairment, are severe health outcomes. Children's education may affect other parental health outcomes or health behaviours, such as adherence to HIV treatment, smoking or salt consumption.[16] In China, for instance, an education programme delivered to schoolchildren as part of the formal school curriculum reduced cardiovascular risk factors among their parents.[35] In Malawi, a large school-based intervention is currently under way to reduce cardiovascular risk factors in children and their family members.[36] Non-communicable diseases, in particular, are expected to pose an increasing challenge in Southern Africa, where health systems have focused on tackling infectious diseases and maternal and child health.[37] Seventh, we observed exposed children only up to age 30 years. Parents' survival and disability may not have been affected during this time period. However, this is a common limitation of prevention studies and our analysis captures longer follow-up than randomised controlled trials which observe parental health over a shorter (eg, 1–2 years) horizon.[35] Eighth, the census data only provided information on our secondary parental outcomes for the subset of children living with at least one parent. The effect of children's education on parental disability, for instance, may differ for children and parents who are not living under the same roof. Additionally, selection bias may occur if the decision of children to coreside with parents depends on exposure to the reform and/or our outcomes for parental disability. However, our estimates for parental disability would only be biased by selection if the reform affected parental disability (ie, our estimates are likely to be a valid test of the null hypothesis that offspring education does not affect their parents' health, even if there is selection bias).[38] Further investigation is warranted to examine if—and how—involving children in their parents' care may improve the health of the older generation in the region.

Overall, our findings suggest that increasing secondary education of offspring may play a more muted role in the survival and disability of parents than suggested by non-experimental observational and longitudinal studies in LMICs. The lack of parental health gains weakens the evidence base for large 'upward' intergenerational benefits of human capital investment.

**Author affiliations**
[1]Heidelberg Institute of Global Health, Medical Faculty and University Hospital, Heidelberg University, Heidelberg, Germany
[2]Medical Research Council Integrative Epidemiology Unit, University of Bristol, Bristol, UK
[3]Population Health Sciences, Bristol Medical School, University of Bristol, Bristol, UK
[4]K.G. Jebsen Center for Genetic Epidemiology, Department of Public Health and Nursing, Norwegian University of Science and Technology (NTNU), Trondheim, Norway
[5]Department of Global Health, Boston University School of Public Health, Boston, Massachusetts, USA
[6]Department of Epidemiology, Boston University School of Public Health, Boston, Massachusetts, USA

**Correction notice** This article has been corrected since it first published. The provenance and peer review statement has been included.

**Acknowledgements** We thank the respondents in the Botswana Census 2001 and 2011, and the staff at the Botswana Central Statistics Office. We also thank Prashant Bharadwaj for helpful comments.

**Contributors** JOL and JWDN conceived and designed the study. JOL conducted the statistical analyses under the guidance of NMD, JB and JWDN. JOL wrote the first draft of the report. NMD, JB and JWDN reviewed and contributed important revisions to the report. All authors approved the final submitted version of the report.

**Funding** JOL was supported by the Else Kröner-Fresenius-Stiftung within the Heidelberg Graduate School of Global Health. NMD was supported by an Economics and Social Research Council (ESRC) Future Research Leaders grant (grant number: ES/N000757/1) and a Norwegian Research Council grant (grant number: 295989). JB and JWDN were supported by the NICHD of National Institutes of Health (grant number: R03-HD098982). JWDN was also supported by the Alexander von Humboldt Foundation funded by Germany's Federal Ministry of Education and Research Health, German Research Foundation (grant number: 405898232) and Heidelberg University's Excellence Initiative. The Medical Research Council (MRC) and the University of Bristol support the MRC Integrative Epidemiology Unit (MC_UU_00011/1).

**Disclaimer** The funders had no role in study design, data collection and analysis, decision to publish, or preparation of the manuscript. The contents are the responsibility of the authors and do not necessarily reflect the views of any of the funders or the US government.

**Competing interests** None declared.

**Patient consent for publication** Not required.

**Ethics approval** This study was preregistered and approved by the Heidelberg University Hospital Ethics Committee (S-186/2018).

**Provenance and peer review** Not commissioned; externally peer reviewed.

**Data availability statement** Data are publicly available at no cost from IPUMS (http://ipums.org).

**ORCID iD**
Jan-Walter De Neve http://orcid.org/0000-0003-0090-8249

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
