## [Reviewer comments · BMJ Open]

ARTICLE DETAILS

TITLE (PROVISIONAL)	Causal Effect of Children's Secondary Education on Parental Health Outcomes – Findings from a Natural Experiment in Botswana
AUTHORS	Ludwig, Jan Ole; Davies, Neil; Bor, Jacob; De Neve, JW

VERSION 1 – REVIEW

REVIEWER	Yitagesu Habtu Addis Ababa University
REVIEW RETURNED	01-Sep-2020

GENERAL COMMENTS	Reviewer comments Title: "Causal Effect of Children's Education on Parental Health– Findings from a Natural Experiment in Botswana" The manuscript assesses the effect of the 1996 Botswana's education policy reform of children's secondary schooling on parental survival from the 2001 and 2011 Botswana decennial census. The paper may have significance if the following points are clearly entertained: • Would have been better if "Parental Health" in the title, changed to "Parental Health Outcome"• The statement describing the result in the abstract section "The 1996 reform caused a large increase in grade ten enrollment, inducing an additional 0.4 years of schooling for the first cohorts affected" seems to have confusion. Is that the study's outcome (effect of children's secondary education on parental health) as indicated in the topic? In addition, this again appeared in the introduction section of this paper in line from line 18 to 23.• What is the implication of the findings in relation to the "the treatment of interest (the 1996 educational reform" Is that currently in operation? Don't you think that many unknown parameters could have caused biased on results (what standardized elements of the 1996 reform existing to the changes over time) due to the longitudinal nature of the study?• Selection of study participants in Section 2.1 that is based on the diagram of Figure 2, why only those children whose parents of at least 1 are excluded from the study?• What could have been individual attributes collected in the 2001 census matched with the same person the 2011 census. For example, predictors such as "age" and "sex" collected from "child x" in the 2001 census may be similar from the 2011 census in such randomly selected data from the large census. There might be a higher correlation among predictors. This may have a significant impact on the outcome of interest although you clearly state your assumptions. How could you handle such biases, possibly "systematic biases"? Could it be detected in your bias component scale?
--

	 • How the possibility of correlation be handled using ordinary least squares (OLS) linear probability regression models keeping your steps of analysis is very ok? • In section 2.4 (instrumental variable analysis), page 11, from line 37 -39, two models are stated. In the causal link, how you address unobserved factors? Don't you think that these factors are mediators? If so, in addition to explaining the three steps, three models are expected in addition to the models explained inline 37-39 and the total effect of your exposure variable (children's year of schooling) with clearly and concisely stated plotting of mediation equations, and errors/disturbances terms for all models.
--	--

REVIEWER	Maame E. Woode Centre for Health Economics, Monash University, Australia
REVIEW RETURNED	01-Sep-2020

GENERAL COMMENTS	Dear Editor, Thank you for giving me the opportunity to review the paper "Causal Effect of Children's Education on Parental Health – Findings from a Natural Experiment in Botswana." This was an interesting paper to read and I would recommend edits. My comments relate largely to the Materials and Methods section where I feel things can further be improved. Thank you. Comments Materials and Methods  1. The policy being studied is a move of grade 10 from senior secondary to junior secondary schools. Does this grade 10 shift translate into an increase in senior secondary school enrolments and potentially university enrolments?  a. If not, then it seems, based on Figure 1, that there are actually slightly less students graduating with junior secondary certificates compared to before the reform, translating into even fewer children attending both senior secondary and tertiary institutions. 2. How was HIV education carried out in secondary schools and are children able to freely talk to their parents on this topic?  a. What would be interesting is to know the status of the parent before the policy was implemented and their status after the implementation of the policy. b. It would also be helpful to know whether infected parents were more likely to seek ARV treatment the implementation of the policy. Both of these would affect parental survival. More generally, are health behaviours different post policy reform? 3. What were the children who did not attend senior secondary school generally spend their time on? If it is work, then household incomes are also affected with the addition of an extra year to junior secondary school education. 4. How do you determine financial transfers from children to parents if they co-reside? 5. You look at the probability that a child will complete at least 10 years of schooling. Will moving grade 10 to junior secondary not imply an increase in probability of completing 10 years of schooling because this becomes the new requirement to obtain a junior certificate?  a. While the policy implies more people are enrolling in grade 10, they still come out with the same certificate they would have obtained under the old policy.
---

	b. If the reform is to have an effect on education/enrolment beyond extra knowledge gained then it would be worthwhile looking at its impact on enrolment in senior secondary school and even in tertiary institutions. 6. What are the first stage results for the 2SLS model? 7. Assuming that the exposure to the policy only affected parental outcomes through changes in secondary schooling is a very strong assumption. Household incomes are also affected both at the time of policy implementation and later on. Discussion 8. The conclusion I draw from the analysis is that an additional year spent in junior secondary school does not lead to any detectable improvements in parental survival and disability. The OLS models however not only look at junior secondary education but also senior secondary and tertiary education, something not directly accounted for by the ITT model. 9. In addition, if the children still graduate with the same junior secondary certificate, with no guarantee of furthering their education irrespective of how long they spend in junior secondary school, then their job prospects are likely to not differ by much, implying the income pathway from child education to parental survival is not strong. 10. If the policy does not lead to more children obtaining a higher certificate than they otherwise would have then it is not likely to affect parental survival
--	--

VERSION 1 – AUTHOR RESPONSE

Reviewer #1

Title: “Causal Effect of Children’s Education on Parental Health–Findings from a Natural Experiment in Botswana”. The manuscript assesses the effect of the 1996 Botswana’s education policy reform of children’s secondary schooling on parental survival from the 2001 and 2011 Botswana decennial census. The paper may have significance if the following points are clearly entertained.

Authors’ response:

We thank the Reviewer for these comments.

Would have been better if “Parental Health” in the title, changed to “Parental Health Outcome”

Authors’ response:

As requested by the Reviewer, we have now changed the title to “Parental Health Outcomes”.

The statement describing the result in the abstract section “The 1996 reform caused a large increase in grade ten enrollment, inducing an additional 0.4 years of schooling for the first cohorts affected” seems to have confusion. Is that the study’s outcome (effect of children’s secondary education on parental health) as indicated in the topic? In addition, this again appeared in the introduction section of this paper in line from line 18 to 23.

Authors' response:

When analyzing a policy change as a natural experiment, it is important to first show that the policy effectively shifted the key exposure of interest. The effect of the 1996 reform on children's educational attainment is an important result. This finding fulfils a key condition which underlies our identification strategy (i.e., the relevance condition). Specifically, to interpret our results as causal estimates, the instrumental variable (exposure to the 1996 reform) must be correlated with the endogenous variable (children's educational attainment). We find large effects. The 1996 reform induced an additional 0.4 years of schooling among exposed cohorts (Table 2). This is a substantial effect compared to similar studies using e.g., the raising of school leaving age in the United Kingdom (Davies et al., 2018). We describe this outcome and its relevance for our study in more detail in Sections 2.4 and 2.5.

"Table 2. First stage results: effect of the reform on children's educational attainment."

Reference:

Davies NM, Dickson M, Davey Smith G, van den Berg GJ, Windmeijer F (2018). The Causal Effects of Education on Health Outcomes in the UK Biobank. *Nature Human Behavior*;2(2):117-125.

What is the implication of the findings in relation to "the treatment of interest (the 1996 educational reform)" Is that currently in operation?

Authors' response:

Yes, the 1996 policy change to an education system with three years of junior secondary school is currently still in operation. We have now clarified this in our descriptions of the reform (Text S1).

"The 1996 policy change to an education system with three years of junior secondary school is currently still in operation." (Text S1, supplementary materials)

Don't you think that many unknown parameters could have caused biased on results (what standardized elements of the 1996 reform existing to the changes over time) due to the longitudinal nature of the study?

Authors' response:

A key assumption underlying our model is that the timing of children's birth around the reform period is random conditional on covariates (i.e., no confounding assumption). There are indeed many unknown parameters which may induce changes in parental health outcomes across children's birth cohorts – such as changes in public health programming or socio-economic context. These changes, however, tend to be gradual over time and/or affect children of different ages. These phenomena would result in gradual changes in parental health outcomes across children's ages and would be picked up in our controls for underlying trends in children's age. The availability of multiple census years also allowed us to generate variation in children's age in a given birth cohort. Our models are therefore robust to period effects, which we controlled for implicitly by simultaneously adjusting for children's age and year of birth. We provide additional details on the no confounding condition in Section 2.5 and in the Discussion section. Although we are confident that this condition is met in our instrumental variable analysis, we are aware that this ultimately cannot be empirically proven and acknowledge so in the Discussion section.

“2.5. Assumptions for Causal Inference” (p. 12, Methods section, revised manuscript)

“This implies that children born before and after the reform cutoff were similar, after controlling for children’s single-year age indicators, a continuous term in children’s year of birth, and children’s district of birth. Critically, the availability of multiple census years allowed us to generate variation in children’s age in a given birth cohort group. Our models are therefore also robust to the existence of period effects, which we control for by simultaneously adjusting for children’s age and a continuous term in year of birth (noting that $\text{census_year} = \text{birth cohort} + \text{age}$). We also assessed alternative specifications of children’s age (quadratic, cubic, quartic) and year of birth (quadratic), and restricted the sample to a narrower birth cohort window around the reform cutoff (+/- 3 years and +/- 5 years around the cutoff). We also allowed for different linear slopes in year of birth on either side of the reform cutoff akin to a ‘fuzzy’ regression discontinuity design (23). We also used covariate balance plots to estimate the association of our exposure (schooling) and our instrument (reform indicator) with observed covariates, as described in detail elsewhere (24, 25). While this analysis cannot prove that our instrument is not associated with unobserved factors, it can provide some indication if our instrument is less strongly associated with measured factors than our exposure.” (p.13, Methods section, revised manuscript)

Selection of study participants in Section 2.1 that is based on the diagram of Figure 2, why only those children whose parents of at least 1 are excluded from the study?

Authors’ response:

We employed the following exclusion criteria for our primary analysis: (i) no information available on our main variables of interest; (ii) children not citizens of or not born in Botswana; and (iii) children who were younger than 18 years or born before 1975 (Section 2.1). These exclusion criteria yielded a final analytical sample of 89,721 children with data on parental survival (Table 1). Among those, 29,266 children co-resided with at least one parent (Figure 2). The criterion “only those children whose parents of at least 1”, however, as noted by the Reviewer, was not an exclusion criterion in our study.

“Table 1. Selected characteristics of estimation sample.”

“Figure 2. Study participant flow diagram”

What could have been individual attributes collected in the 2001 census matched with the same person the 2011 census. For example, predictors such as “age” and “sex” collected from “child x” in the 2001 census may be similar from the 2011 census in such randomly selected data from the large census. There might be a higher correlation among predictors. This may have a significant impact on the outcome of interest although you clearly state your assumptions. How could you handle such biases, possibly “systematic biases”? Could it be detected in your bias component scale? How the possibility of correlation be handled using ordinary least squares (OLS) linear probability regression models keeping your steps of analysis is very ok?

Authors’ response:

In our study, we used pooled cross-sectional data from the decennial censuses (2001 and 2011). Our census dataset, however, does not allow matching individuals between the two census years. We have now clarified this point in the Methods section of the revised manuscript. Our results are also consistent across different data sources, including using either a 10% random sample of the full census provided by IPUMS or full census itself which is available upon request from the Botswana Central Statistics Office.

“The census data do not allow matching individuals between census years.” (p.7, Methods section, revised manuscript)

In section 2.4 (instrumental variable analysis), page 11, from line 37 -39, two models are stated. In the causal link, how you address unobserved factors? Don't you think that these factors are mediators? If so, in addition to explaining the three steps, three models are expected in addition to the models explained in line 37-39 and the total effect of your exposure variable (children's year of schooling) with clearly and concisely stated plotting of mediation equations, and errors/disturbances terms for all models.

Authors' response:

We thank the Reviewer for this excellent comment. For a causal interpretation of our effect estimates, the no confounding assumption must be satisfied. This assumption suggests that the instrument must be independent of unobserved factors of the instrument-outcome association conditional on observed factors. We discuss the no confounding assumption in Section 2.5 and in the Discussion section. Although we are confident that the no confounding condition is met in our analysis, we are aware that this ultimately cannot be empirically proven and acknowledge so in the Discussion section. Additionally, we now show all three models with error terms as recommended by the Reviewer (p.11, Methods section).

“2.5. Assumptions for Causal Inference” (p. 12, Methods section, revised manuscript)

“This implies that children born before and after the reform cutoff were similar, after controlling for children's single-year age indicators, a continuous term in children's year of birth, and children's district of birth. Critically, the availability of multiple census years allowed us to generate variation in children's age in a given birth cohort group. Our models are therefore also robust to the existence of period effects, which we control for by simultaneously adjusting for children's age and a continuous term in year of birth (noting that census_year = birth cohort + age). We also assessed alternative specifications of children's age (quadratic, cubic, quartic) and year of birth (quadratic), and restricted the sample to a narrower birth cohort window around the reform cutoff (+/- 3 years and +/- 5 years around the cutoff). We also allowed for different linear slopes in year of birth on either side of the reform cutoff akin to a 'fuzzy' regression discontinuity design (23). We also used covariate balance plots to estimate the association of our exposure (schooling) and our instrument (reform indicator) with observed covariates, as described in detail elsewhere (24, 25). While this analysis cannot prove that our instrument is not associated with unobserved factors, it can provide some indication if our instrument is less strongly associated with measured factors than our exposure.” (p.13, Methods section, revised manuscript)

“We estimated first stage, ITT, and 2SLS models of the form:

$$(1) \quad Children'sEduc_i = \beta_0 + \beta_1 1[Children'sYearofBirth > 1980]_i + \beta_2 f(Children'sYearof Birth)_i + \beta_3 1[Children'sAge]_i + \beta_4 1[Children'sDistrictofBirth]_i + \epsilon_i$$

$$(2) \quad ParentalSurvival_i = \gamma_0 + \gamma_1 1[Children'sYearofBirth > 1980]_i + \gamma_2 f(Children'sYearof Birth)_i + \gamma_3 1[Children'sAge]_i + \gamma_4 1[Children'sDistrictofBirth]_i + v_i$$

$$(3) \quad ParentalSurvival_i = \alpha_0 + \alpha_1 Children'sEduc_hat_i + \alpha_2 f(Children'sYearofBirth)_i + \alpha_3 1[Children'sAge]_i + \alpha_4 1[Children'sDistrictofBirth]_i + \zeta_i$$

with Children's Education predicted in (1)." (p.11, Methods section, revised manuscript)

Reviewer #2

Thank you for giving me the opportunity to review the paper "Causal Effect of Children's Education on Parental Health – Findings from a Natural Experiment in Botswana." This was an interesting paper to read and I would recommend edits. My comments relate largely to the Materials and Methods section where I feel things can further be improved. Thank you.

Authors' response:

We thank the Reviewer for these comments.

Materials and Methods. 1. The policy being studied is a move of grade 10 from senior secondary to junior secondary schools. Does this grade 10 shift translate into an increase in senior secondary school enrolments and potentially university enrolments? If not, then it seems, based on Figure 1, that there are actually slightly less students graduating with junior secondary certificates compared to before the reform, translating into even fewer children attending both senior secondary and tertiary institutions.

Authors' response:

We thank the Reviewer for this excellent comment. In prior work, published in *The Lancet Global Health*, we assessed the effects of the policy reform on the probabilities of completing at least 10, 11, 12, and 13+ years of schooling in multivariable OLS regression models. We found modest increases in the completion of at least 11 and 12 years of schooling for the reform cohorts. Specifically, exposure to the policy reform was associated with an increase of 8.2 percentage points ($p < 0.01$) in the completion of senior secondary school (Table R1). The policy reform may have induced some students to enroll in senior secondary school if they, for instance, post-reform realized that they like secondary school.

Reference:

De Neve JW, Fink G, Subramanian SV, Moyo S, Bor, J (2015). Length of secondary schooling and risk of HIV infection in Botswana: evidence from a natural experiment. *Lancet Global Health*, 3, e470-e477.

Table R1: First stage regressions: effect of the policy reform on years of schooling

Coefficient on Reform Indicator	(1) Female	(2) Male	(3) Both Sexes	(4) Both Sexes
Dependent Variable				
Years of Schooling	0.635*** (0.223)	1.005*** (0.322)	0.792*** (0.188)	

At Least 10 Years of Schooling	0.249*** (0.024)
At Least 11 Years of Schooling	0.069*** (0.026)
At Least 12 Years of Schooling	0.082*** (0.026)
At Least 13 Years of Schooling	0.031 (0.020)

Regressions include age dummies, a linear term for year of birth, an indicator for survey wave and dummies for district of birth. Columns 3 and 4 additionally control for age*sex, districtofbirth*sex, yearofbirth*sex and surveywave*sex interactions. Column 4 additionally controls for indicators for at least 7, 8 or 9 completed years of schooling. Standard errors in parentheses. *** p<0.01, ** p<0.05, * p<0.1. No weights used. Sample includes survey respondents who were citizens of Botswana, at least 18 years old at the time of the surveys, born in or after 1975, and had a valid HIV test result. Source: Botswana AIDS Impact Survey II (2004) and III (2008) (N=7,018). Data from De Neve et al. 2015.

2. How was HIV education carried out in secondary schools and are children able to freely talk to their parents on this topic?

Authors' response:

The reform cohorts in our study completed secondary school in the 1990s and early 2000s, before Botswana launched a formal HIV curriculum in schools (Government of Botswana, 1994). We also examined norms and attitudes around discussing HIV with others (e.g., spouse, family members, friends, health workers) in prior work using detailed data from the Botswana AIDS Impact Surveys of 2004 and 2008 (De Neve & Bor 2015). Specifically, we found that 47.1% of respondents “discussed HIV/AIDS with anyone in the past 4 weeks”. However, most respondents did not discuss HIV with family members (e.g., older parents) but rather with friends. Consistent with our main results in the current study, these findings suggest that children’s exposure to the reform was unlikely to lead to substantial upward transfers in HIV knowledge, from children to parents, nor discussions around HIV with parents.

Reference:

Bor J, De Neve JW (2015). A Social Vaccine? HIV Infection, Fertility, and the Non-Pecuniary Returns to Secondary Schooling in Botswana. Population Association of America Annual Meeting. San Diego. Available at: <https://paa2015.princeton.edu/abstracts/151006>.

What would be interesting is to know the status of the parent before the policy was implemented and their status after the implementation of the policy. It would also be helpful to know whether infected parents were more likely to seek ARV treatment the implementation of the policy. Both of these would affect parental survival. More generally, are health behaviors different post policy reform?

Authors' response:

In our study, we compare the survival and disability status of parents of children born pre- and post-reform using pooled cross-sectional data from the decennial census of Botswana (2001 and 2011). However, the census data do not contain information enabling linkage of data on the same parents over time nor on parental health behaviors (e.g., HIV testing, ARV treatment). One reason to conduct the current study was to see whether it would be useful to include questions on upward spillover effects from children's education to their parents in future surveys in HIV endemic settings. Our results, however, suggest that children's secondary education may have few detectable effects on parental survival and disability in this context.

3. What were the children who did not attend senior secondary school generally spend their time on? If it is work, then household incomes are also affected with the addition of an extra year to junior secondary school education.

Authors' response:

This is a great point. Conceptually, people who don't stay in school might gain advantages in work experience leading to higher wages. On the other hand, schooling could also increase wages and employment opportunities. Our analysis allows for both impacts, assessing the impact of schooling on parental health through these and myriad other pathways.

In fact, previous studies have found that the policy reform substantially improved economic outcomes, including increased labor force participation (Borkum, 2009; Bor & De Neve, 2015), skills (e.g., literacy) (Bor & De Neve, 2015), occupational skill levels (Lindskog & Durevall 2019), as well as income (Borkum, 2009) (Text S1). In response to the Reviewer's comment, we replicated these findings for labor force participation using our census dataset. We found that each additional year of schooling induced by the policy reform increased labor force participation by 13.0 percentage points (95% CI: 8.0 – 18.1). The reform caused over half of those women who would have otherwise been out of the labor force to seek employment. For the current study, we hypothesized that these improvements in economic opportunity may have enabled upward transmission of human capital, from children to their parents.

"Text S1. Study context and education policy reform" (Supplementary materials)

References:

Bor J, De Neve JW (2015). A Social Vaccine? HIV Infection, Fertility, and the Non-Pecuniary Returns to Secondary Schooling in Botswana. Population Association of America Annual Meeting. San Diego. Available at: <https://paa2015.princeton.edu/abstracts/151006>

Borkum E (2009) Grade structure, educational attainment and labor market outcomes: Evidence from Botswana. Columbia University (Job Market Paper).

Lindskog A, Durevall D (2019). To educate a woman and to educate a man: Gender-specific sexual behaviour and HIV responses to an education reform in Botswana. Economics Department, University of Gothenburg Working Papers in Economics 763. Available at: <https://gupea.ub.gu.se/handle/2077/60262>

4. How do you determine financial transfers from children to parents if they co-reside?

Authors' response:

We were unable to determine financial transfers between children and parents because this data was unfortunately not collected during the census. Our analysis assesses the impact of schooling through all potential mechanisms, but due to data limitations we cannot investigate the mechanisms that mediate the effects. We have clarified this in the Discussion section, where we write:

“Data on financial transfers between children and their parents ... were not collected during the census.” (p.7, Methods section, revised manuscript)

5. You look at the probability that a child will complete at least 10 years of schooling. Will moving grade 10 to junior secondary not imply an increase in probability of completing 10 years of schooling because this becomes the new requirement to obtain a junior certificate? While the policy implies more people are enrolling in grade 10, they still come out with the same certificate they would have obtained under the old policy. If the reform is to have an effect on education/enrolment beyond extra knowledge gained then it would be worthwhile looking at its impact on enrolment in senior secondary school and even in tertiary institutions.

Authors' response:

Yes, the policy reform raised the benefits of completing grade 10 because it was now required to obtain a Junior Certificate (Text S1, supplementary materials). Additionally, we assessed the effects of the reform on the probabilities of completing at least 10, 11, 12, and 13+ years of schooling in multivariable OLS regression models reported in detail elsewhere (De Neve et al., 2015). We found modest increases in the completion of at least 11 and 12 years of schooling for the reform cohorts. Specifically, exposure to the policy reform was associated with an increase of 8.2 percentage points ($p < 0.01$) in the completion of senior secondary school (Table R1). The policy reform may have induced some students to enroll in senior secondary school if they, for instance, post-reform realized that they like secondary school. This may result in positive bias (i.e. suggesting that the policy reform did effect parental health). However, because we find few detectable effects of children's education on parental health, it appears unlikely that this source of bias would bias our results towards the null.

Reference:

De Neve JW, Fink G, Subramanian SV, Moyo S, Bor, J (2015). Length of secondary schooling and risk of HIV infection in Botswana: evidence from a natural experiment. *Lancet Global Health*, 3, e470-e477.

What are the first stage results for the 2SLS model?

Authors' response:

The first stage results for the 2SLS model are presented in Table 2. We show the estimated reform impact on children's educational attainment, as indicated by total years of schooling completed and the probability of having completed at least 10 years of schooling. We find large effects: in the pooled sample, the estimated effect of the reform on educational attainment was 0.4 additional years of schooling (95% CI: 0.3 – 0.5) among exposed child cohorts. Exposure to the reform was also associated with a 19.0 percentage point increase (95% CI: 17.5 – 20.4) in the probability of having completed at least ten years of schooling relative to a baseline probability of 43.5% among pre-reform child cohorts.

“Table 2. First stage results: effect of the reform on children's educational attainment.”

“First, our instrument must be associated with educational attainment. This is testable and we found evidence of large effects (F-statistic: 65.6, Table 3). The large value of the F-statistic indicates that our instrument was strongly associated with schooling.” (p.12, Methods section, revised manuscript)

7. Assuming that the exposure to the policy only affected parental outcomes through changes in secondary schooling is a very strong assumption. Household incomes are also affected both at the time of policy implementation and later on.

Authors’ response:

We assume that exposure to the policy reform affected our parental outcomes only through changes in secondary schooling (Section 2.4). This assumption is plausible given that the education reform was a supply-side intervention (Government of Botswana, 1994), which would not have affected parents except through their children’s increased access to school. Further, we use the legal structure of the policy change as our instrument for the actual changes in schooling observed, since it is the legal changes that are exogenous to children’s choices (and their parents) about schooling. Revealed choices about educational attainment, in contrast, may be correlated with other determinants of e.g., income.

The reviewer suggests that household incomes could be affected by the policy reform and could affect later outcomes. We see this as part of the exposure of interest. The opportunity costs of schooling (not being able to work) are part and parcel of schooling, and it is impossible to increase schooling without reducing whatever activities a person would have been doing at the time. Therefore, the exposure could be most precisely stated as ‘an additional year of schooling – and a year less of whatever the child would have been doing otherwise’. We have now further clarified this point in the Discussion section:

“Fourth, the opportunity costs of schooling (e.g., not being able to work) are part and parcel of schooling, and it is impossible to increase schooling without reducing whatever activities children would have been doing at the time. Therefore, the exposure could be most precisely stated as an additional year of schooling – and a year less of whatever the child would have been doing otherwise.” (p.20, Discussion section, revised manuscript)

Reference:

Government of Botswana. The revised national policy on education. Gaborone, Botswana: Botswana Government Printer; 1994.

Discussion. 8. The conclusion I draw from the analysis is that an additional year spent in junior secondary school does not lead to any detectable improvements in parental survival and disability. The OLS models however not only look at junior secondary education but also senior secondary and tertiary education, something not directly accounted for by the ITT model.

Authors’ response:

We fully agree with the Reviewer. The variation explored in this paper mostly reflects variation in junior secondary schooling. Effects of schooling, however, may be different for primary (e.g., De Neve & Fink 2018) or tertiary education (e.g., Chen et al., 2019). We have acknowledged this limitation in the Discussion section. Further, in Figure 3, we illustrate heterogeneity by school level in the naïve

relationship between children’s education and parental health. We find that the association between children’s education and parental survival was non-linear, levelling off after junior secondary school.

“While the educational reform likely allows us to identify the causal effects of interest, relying on changes generated by the 1996 education reform and census data comes with limitations in terms of external validity. First, all of the variation explored reflects variation in secondary schooling. Effects of schooling may be qualitatively and quantitatively different in primary school (30, 31) or higher education (10), and indeed the association between measured schooling and parental survival was non-linear. Second, the causal effects we identify do not necessarily represent the causal effects in the general population but in the subpopulation of compliers (i.e. children who attained increased schooling because of the reform). Nevertheless, one advantage of using Botswana’s supply-side school reform is the important size of its impact (nearly half a year of additional schooling on a population level), and that it affected a large segment of the population.” (p. 20, Discussion section, revised manuscript)

“Figure 3. Parental survival and disability by children’s educational attainment.”

Notes: Figure 3 shows the unadjusted relationship between children’s schooling and parental survival status and disability. Educational attainment was defined as total years of schooling completed by the time of the census. Strong protective effects are seen for primary school completion, with effects levelling off after junior secondary school. The sample for parental survival includes respondents who are Botswana citizens, born in Botswana, and ages at least 22 at the time of the census. The sample for parental disability includes respondents who are Botswana citizens, born in Botswana, ages at least 22, and living with at least one parent at the time of the census. Source: Botswana Census 2011.

References:

Chen Y, Persson P, Polyakova M (2019). The Roots of Health Inequality and The Value of Intra-Family Expertise. National Bureau of Economic Research (NBER) Working Paper No. 25618; [Available at: <https://www.nber.org/papers/w25618>].

De Neve JW, Fink G (2018). Children's education and parental old age survival - Quasi-experimental evidence on the intergenerational effects of human capital investment. *J Health Econ*; Mar;58:76-89.

9. In addition, if the children still graduate with the same junior secondary certificate, with no guarantee of furthering their education irrespective of how long they spend in junior secondary school, then their job prospects are likely to not differ by much, implying the income pathway from child education to parental survival is not strong.

Authors' response:

Additional years of schooling may increase wages by increasing people's skills (e.g. literacy, numeracy) even without additional credentials. Previous studies have found that the policy reform substantially improved economic outcomes, including increased labor force participation (Borkum, 2009; Bor & De Neve, 2015), skills (e.g., literacy) (Bor & De Neve, 2015), occupational skill levels (Lindskog & Durevall 2019), as well as income (Borkum, 2009) (Text S1). In response to the Reviewer's comment, we replicated these findings for labor force participation using our census dataset. We found that each additional year of schooling increased labor force participation by 13.0 percentage points (95% CI: 8.0 – 18.1). The reform caused over half of those women who would have otherwise been out of the labor force to seek employment. Overall, this suggests that the reform had substantial positive effects on economic outcomes (Text S1 in the supplementary materials), but little effect on parental survival or disability. For the current study, we hypothesized that these improvements in economic opportunity may have enabled upward transmission of human capital, from children to their parents.

"Text S1. Study context and education policy reform" (Supplementary materials)

References:

Bor J, De Neve JW (2015). A Social Vaccine? HIV Infection, Fertility, and the Non-Pecuniary Returns to Secondary Schooling in Botswana. Population Association of America Annual Meeting. San Diego. Available at: <https://paa2015.princeton.edu/abstracts/151006>

Borkum E (2009) Grade structure, educational attainment and labor market outcomes: Evidence from Botswana. Columbia University (Job Market Paper).

Lindskog A, Durevall D (2019). To educate a woman and to educate a man: Gender-specific sexual behaviour and HIV responses to an education reform in Botswana. Economics Department, University of Gothenburg Working Papers in Economics 763. Available at: <https://gupea.ub.gu.se/handle/2077/60262>

10. If the policy does not lead to more children obtaining a higher certificate than they otherwise would have then it is not likely to affect parental survival.

Authors' response:

Additional years of schooling may increase wages or improve health by increasing people's skills (e.g. literacy, numeracy) even without additional credentials. Additionally, while sheepskin effects are well documented for wages, there is relatively little evidence of sheepskin effects in health (Cutler and Lleras-Muney, 2008). Prior to the policy reform, the Botswana National Commission on Education had criticized the former "7-2-3" system for not adequately preparing students for the labor market (see Text S1 in the supplementary materials for additional discussion on the study context and policy reform). After the policy reform, it was required to complete an additional year (Grade 10) to obtain a Junior Certificate, which had substantial economic benefits. In our census dataset, each additional year of schooling resulting from the policy reform increased labor force participation by 13.0 percentage points (95% CI: 8.0 – 18.1). The reform also increased literacy skills (Bor & De Neve, 2015), occupational skill levels (Lindskog & Durevall 2019), as well as income (Borkum, 2009). In this study, we hypothesized that these changes in economic opportunity may have enabled upward transmission of human capital, from children to their parents. These economic outcomes also provide good positive "controls". The reform had the economic effects we would expect at an individual level, but there were few detectable effects on parent's health outcomes. This is consistent with children having limited effects on their parents' health.

Reference:

Cutler D, Lleras-Muney A (2008). "Education and Health: Evaluating Theories and Evidence." Making Americans Healthier: Social and Economic Policy as Health Policy, edited by J House, R Schoeni, G Kaplan, and H Pollack. New York: Russell Sage Foundation.

VERSION 2 – REVIEW

REVIEWER	Yitagesu Habtu Addis Ababa University Addis Ababa, Ethiopia
REVIEW RETURNED	02-Nov-2020

GENERAL COMMENTS	All comments are addressed. I would have preferred if the author had made agreement on the following two statements. 1. "The 1996 reform caused a large increase in grade ten enrollment, inducing an additional 0.4 years of schooling for the first cohorts affected" in abstract section (Abstract section) 2. "At the cohort level, the reform induced an additional 0.4 years of schooling among affected cohorts relative to the attainment the affected cohorts would have achieved if previous trends had continued" (Introduction section).
---

REVIEWER	Maame Esi Woode Centre for Health Economics, Monash Business School, Monash University, Australia
REVIEW RETURNED	19-Nov-2020

GENERAL COMMENTS	My questions have been answered and while I feel supplementary appendix S1 will be better off in the main text, I understand that word count limits makes this impossible.
--